# Nasty Adversarial Training: A Probability Sparsity Perspective for Robustness Enhancement

**Yuhang Zhou**
School of Computer Science and Technology
Harbin Institute of Technology, Shenzhen
Shenzhen, China
{23B951015}@stu.hit.edu.cn

**Zhongyun Hua**
School of Computer Science and Technology
Harbin Institute of Technology, Shenzhen
Shenzhen, China
{huazhongyun}@hit.edu.cn

**Zhaoquan Gu**
School of Computer Science and Technology
Harbin Institute of Technology, Shenzhen
Peng Cheng Laboratory
Shenzhen, China
{guzhaoquan}@hit.edu.cn

**Keke Tang**
Cyberspace Institute of Advanced Technology
Guangzhou University
Guangzhou, China
{tangbohutbh}@gmail.com

**Rushi Lan**
School of Computer Science and Information Security
Guilin University of Electronic Technology
Guilin, China
{rslan2016}@163.com

**Yushu Zhang**
School of Computing and Artificial Intelligence
Jiangxi University of Finance and Economics
Nanchang, China
{yushu}@nuaa.edu.cn

**Qing Liao**
School of Computer Science and Technology
Harbin Institute of Technology, Shenzhen
Shenzhen, China
{liaoqing}@hit.edu.cn

**Leo Yu Zhang**
School of Information and Communication Technology
Griffith University
Brisbane, Australia
{leo.zhang}@griffith.edu.au

## Abstract

The vulnerability of deep neural networks to adversarial examples poses significant challenges to their reliable deployment. Among empirical defenses, adversarial training and robust distillation remain the most effective. In this paper, we identify a property originally studied in the context of model intellectual property protection, i.e., probability sparsity induced by nasty training, and reveal its potential to enhance adversarial robustness in an interpretable manner. We first analyze how nasty training drives models toward sparse probability distributions and qualitatively explore the spatial metric preferences introduced by such sparsity. Building on these insights, we propose nasty adversarial training (NAT), a simple yet effective adversarial training framework that incorporates probability sparsity as a regularization mechanism to strengthen robustness. Both theoretical analysis and extensive experiments demonstrate the effectiveness of NAT, showing that probability sparsity not only improves adversarial resilience but also provides interpretability to the robustness gains.

## 1 Introduction

Despite their success in a wide range of computer vision tasks, deep neural networks (DNNs) remain highly vulnerable to adversarial attacks, raising serious challenges to their reliability and secure deployment. By introducing carefully crafted perturbations to input, adversaries can easily mislead models into producing incorrect predictions. As these perturbations are often imperceptible

or barely noticeable to the human eye, such vulnerabilities pose significant risks in safety-critical applications such as autonomous driving, where manipulated traffic signs may cause catastrophic accidents (Eykholt et al., 2018).

Adversarial defense aims to preserve robustness under such attacks. Although early defense methods (Xie et al., 2019a; Liao et al., 2018; Papernot et al., 2016; Xie et al., 2017a) reported promising robustness, they were later shown to rely on "obfuscated gradients" (Athalye et al., 2018) and proved ineffective against adaptive adversaries. Currently, the most reliable empirical defenses are adversarial training (AT) (Madry et al., 2017; Jia et al., 2022a; Zhou & Hua, 2024; Zhang et al., 2024a; Yu-Hang et al., 2025) and robustness distillation (RD) (Zhu et al., 2021; Zi et al., 2021; Goldblum et al., 2020; Zhou et al.). AT improves robustness by dynamically generating adversarial examples during training, while RD extends this paradigm with improved inference efficiency, making it suitable for deployment on resources-constrained platforms such as edge devices.

Although AT has many variants, few studies have explored robustness attribution from the perspective of the model's output probability distribution. Nasty training (NT) (Ma et al., 2021), which regularizes models using a nasty adversary, provides an intriguing perspective by suggesting that probability sparsity may also contribute to adversarial robustness. The original goal of NT is to train a teacher model resistant to imitation by a student model. Subsequent work by Ma et al. (2022) theoretically linked this effect to probability sparsity, showing that sparse probability distributions can hinder distillation. However, NT does not explicitly address adversarial robustness and largely overlooks the potential impact of probability sparsity in this context. We argue that probability sparsity reflects greater inter-class separation in the output logits, offering a spatial-metric interpretation of robustness. Employing this insight could unlock additional potential in adversarial training. Additional detailed related work is provided in the appendix.

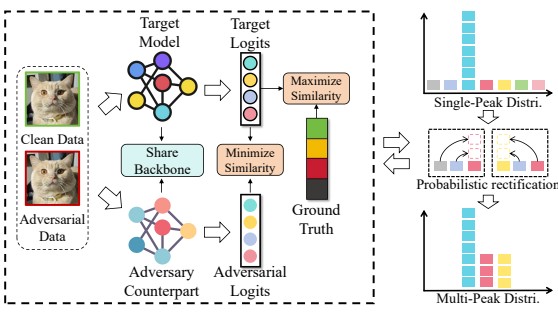

Figure 1: Framework of the proposed NAT. NAT enhances the target model's training by introducing an auxiliary adversary model in addition to the primary training loss. Probability sparsity regularization (i.e., probabilistic rectification) is achieved by maximizing the divergence between the output distributions of the target model and the adversary model.

In this paper, we first investigate the underlying cause of sparse probability outputs in NT, attributing it to high-order power optimization. We then qualitatively analyze the spatial metric properties induced by these sparse probability distributions, including enhanced class separability and larger attack tolerance in the classification layer. Motivated on these insights, we propose nasty adversarial training (NAT), a simple yet efficient adversarial defense that utilizes probability sparsity as a regularization mechanism to enhance adversarial robustness. NAT introduces an auxiliary adversary model that applies nasty regularization alongside the primary training objective. The target model achieves probability sparsity by maximizing the divergence between its outputs and those of the adversary model, while simultaneously learning discriminative ability by minimizing divergence with the ground truth. Experimental experiments demonstrate that NAT achieves state-of-the-art adversarial robustness with minimal overhead. Our main contributions are summarized as follows:

(1) We analyze the attribution of probability sparsity in NT and reveal its spatial metric benefits, providing interpretable insights into adversarial robustness.

(2) We propose NAT, a novel adversarial defense that integrates NT as an effective AT regularizer.

(3) We empirically validate that NAT achieves state-of-the-art adversarial robustness with low computational overhead. Ablation studies and discussions further highlight its effectiveness.

## 2 METHODOLOGY

### 2.1 PRELIMINARY

**Nasty Training (NT).** The original objective of NT is to train a target teacher model $f_{\theta_t}$ that resists distillation by a student model $f_{\theta_s}$, where $\theta_t$ and $\theta_s$ represent their respective model parameters. In standard knowledge distillation, the student model is trained to approximate the outputs of the teacher model. This process can be formalized as:

$$
\min_{\theta_S} \sum_{(x_i, y_i) \in \mathcal{X}} \alpha \tau_s^2 \mathrm{KL} \left( \sigma_{\tau_s} \left( p_{f_{\theta_T}} (x_i) \right), \sigma_{\tau_s} \left( p_{f_{\theta_S}} (x_i) \right) \right)
$$
$$
+ (1 - \alpha) \mathrm{XE} \left( \sigma \left( p_{f_{\theta_S}} (x_i) \right), y_i \right),
\tag{1}
$$

where $\mathrm{KL}(\cdot)$ and $\mathrm{XE}(\cdot)$ denote the KL-divergence and cross-entropy loss, respectively. The hyper-parameter $\alpha$ balances the trade-off between distillation and classification objectives. $\sigma_{\tau_s}(\cdot)$ denotes the "softmax temperature" function (Ma et al., 2021), which reduces to the standard softmax $\sigma(\cdot)$ when $\tau_s = 1$. Here, $\tau_s$ and $\tau_a$ are softmax temperatures, $\mathcal{X}$ is the target distribution, and $p$ represents the output probabilities.

In NT, to prevent the teacher model from being distilled by the student model, an auxiliary adversary model $f_{\theta_a}$, which is a vanilla-trained counterpart of the target model in the original setting, is introduced. This adversary model enables the target teacher model to learn task-relevant knowledge and maximize its output discrepancy from the adversary model. This training objective can be formalized as:

$$
\min_{\theta_t} \sum_{(x_i, y_i) \in \mathcal{X}} \mathrm{XE} \left( \sigma \left( p_{f_{\theta_t}} (x_i) \right), y_i \right)
$$
$$
- \omega_a \mathrm{KL} \left( \sigma_{\tau_a} \left( p_{f_{\theta_t}} (x_i) \right), \sigma_{\tau_a} \left( p_{f_{\theta_a}} (x_i) \right) \right),
\tag{2}
$$

where $\omega_a$ is a hyper-parameter that balances the target cross-entropy loss and the nasty loss. Intuitively, the adversary model $f_{\theta_a}$ acts as a surrogate for the student model $f_{\theta_s}$. Instead of allowing the student to distill knowledge by aligning with the teacher, NT reduces the similarity between the teacher and the surrogate adversary's outputs. Subsequently, Ma et al. (2022) further provided a theoretical analysis, showing that the sparsity of the teacher's output probabilities is the key factor that prevents successful distillation.

**Adversarial Attack.** The objective of adversarial attacks is to identify an optimal perturbation $\delta_x$ that causes a well-trained target model $f_{\theta_t}$ to make incorrect predictions on the perturbed input, thereby maximizing the model's loss on the true label. This can be formalized as:

$$
\arg\max_{\delta x} \mathcal{L}(f_{\theta_t}(x + \delta_x), y), \text{ s.t. } ||\delta_x||_p \leqslant \varepsilon,
\tag{3}
$$

where $\varepsilon$ denotes the perturbation budget, $\mathcal{L}$ represents the target loss, typically the cross-entropy loss.

*Adversarial training.* In AT, adversarial examples are dynamically generated during training process and incorporated into the learning process. Through a min-max optimization framework, the model learns to gain adversarial robustness against internal maximization (i.e., adversarial attacks). This process can be formalized as:

$$
\min_{\theta} \mathbb{E}_{(x,y) \sim \mathcal{X}} \left[ \max_{||\delta x||_p \leqslant \varepsilon} \mathcal{L}(x + \delta x, y; \theta) \right].
\tag{4}
$$

### 2.2 NASTY ADVERSARIAL TRAINING

We propose NAT, which incorporates probabilistic sparsity into AT through nasty regularization. Intuitively, NAT extends standard AT by introducing an auxiliary adversary model for the target model, with both models jointly optimized under adversarial training. The overall training objective

is defined as:

$$
\begin{aligned}
\min_{\theta_t} \sum_{(x_i, y_i) \in X \cup X'} & \mathrm{XE}\left(\sigma\left(p_{f_{\theta_t}}(x_i)\right), y_i\right) \\
& - \omega_a \mathrm{KL}\left(\sigma_{\tau_a}\left(p_{f_{\theta_t}}(x_i)\right), \sigma_{\tau_a}\left(p_{f_{\theta_a}}(x_i)\right)\right) \\
\text{s.t. } X' = & \left\{x' \;\middle|\; \arg\max_{x'} \mathrm{XE}\left(\sigma\left(p_{f_{\theta_t}}(x_i)\right), y_i\right), \forall x \in X\right\},
\end{aligned}
\tag{5}
$$

where $X'$ denotes the set of adversarial examples corresponding to the original input distribution. Following the original NT setup, we use a vanilla-trained counterpart of the target teacher as the adversary model. This configuration allows the target model to learn sparse output probability distributions while simultaneously improving adversarial robustness. In the next section, we analyze the origin of sparsity in NT and qualitatively explain how probabilistic sparsity contributes to robustness by shaping spatial relationships in the output space.

# 3 IN-DEPTH ANALYSIS

In this section, we present two key analyses to explain how NAT induces probabilistic sparsity and strengthens robustness: (1) the origin of probabilistic sparsity in NT, and (2) its influence on the model robustness from the perspective of spatial structure.

## 3.1 ATTRIBUTION OF PROBABILITY SPARSITY

Previous work (Ma et al., 2022) primarily focused on the effect of probability sparsity on knowledge distillation, but did not investigate why NT induces such sparsity. In this paper, we further explore the origin of probability sparsity in NT by performing a Taylor expansion of the adversary regularization term. Specifically, beyond the primary classification objective (i.e., cross-entropy loss), we reformulate the adversary regularization term in NT as follows:

$$
\begin{aligned}
\mathcal{L}_{\text{Nasty}} &= -\frac{1}{N}\sum_{i=1}^{N}\sum_{c=1}^{C} q_{i,c}^t \cdot \log\left(\frac{q_{i,c}^t}{q_{i,c}^a}\right) \\
&= -\frac{1}{N}\sum_{i=1}^{N}\sum_{c=1}^{C}\left[q_{i,c}^t \cdot \log\left(q_{i,c}^t\right) - q_{i,c}^t \cdot \log\left(q_{i,c}^a\right)\right].
\end{aligned}
\tag{6}
$$

where $N$ denotes the number of samples and $C$ the number of categories. $q_{i,c}^t$ and $q_{i,c}^a$ are the output logits of the target model and adversary model for the $i_{th}$ example and $c_{th}$ category, respectively. Our analysis focuses on the second term, since the first term is independent of the adversary model and therefore does not capture its effect. Let $\Delta_{i,c}^q = q_{i,c}^a - q_{i,c}^t$, we expand $\log(q_{i,c}^a)$ around $q_{i,c}^t$ using a Taylor series:

$$
\begin{aligned}
\log\left(q_{i,c}^a\right) = \log\left(q_{i,c}^t\right) &+ \frac{1}{q_{i,c}^t}\Delta_{i,c}^q - \frac{1}{2\left(q_{i,c}^t\right)^2}\left(\Delta_{i,c}^q\right)^2 \\
&+ \cdots + \frac{(-1)^{(K+1)}}{K(q_{i,c}^t)}\left(\Delta_{i,c}^q\right)^K
\end{aligned}
\tag{7}
$$

Following the assumption of Ma et al. (2022), we let $q_{i,c}^a = q_{i,c}^t + \Delta_{i,c}^q$, where $\Delta_{i,c}^q$ denotes the gap between $q_{i,c}^a$ and $q_{i,c}^t$. Each of $q_{i,c}^a$, $q_{i,c}^t$, and $\Delta_{i,c}^q$ can be regarded as functions of the input $x_{i,c}$, i.e., $q_{i,c}^a(x_{i,c})$, $q_{i,c}^t(x_{i,c})$, and $\Delta_{i,c}^q(x_{i,c})$. Although $\Delta_{i,c}^q(x_{i,c})$ is abstract and difficult to express explicitly, it induces an implicit functional dependence between the cross-entropy loss and $q_{i,c}^a$, since both $q_{i,c}^a$ and $q_{i,c}^t$ are functions of the input $x_{i,c}$. As a result, the cross-entropy can be rewritten as $L(x_{i,c}, q_{i,c}^a - \Delta_{i,c}^q)$, enabling a Taylor expansion. Based on this formulation, the regularization term

can be approximated as:

$$\mathcal{L}_{Nasty} \approx \frac{1}{N} \sum_{i=1}^{N} \sum_{c=1}^{C} (q_{i,c}^a - q_{i,c}^t) - \frac{1}{2N} \sum_{i=1}^{N} \sum_{c=1}^{C} \frac{(q_{i,c}^a - q_{i,c}^t)^2}{q_{i,c}^t}$$
$$+ \frac{1}{3N} \sum_{i=1}^{N} \sum_{c=1}^{C} \frac{(q_{i,c}^a - q_{i,c}^t)^3}{(q_{i,c}^t)^2} + \cdots \quad (8)$$
$$+ \frac{(-1)^{K+1}}{KN} \sum_{i=1}^{N} \sum_{c=1}^{C} \frac{(q_{i,c}^a - q_{i,c}^t)^K}{(q_{i,c}^t)^{K-1}}.$$

The Taylor expansion of the nasty loss provides an intuitive explanation of the mechanism underlying NAT. Beyond the primary cross-entropy loss, NT introduces a higher-order regularization term. The first-order term, $\frac{1}{N} \sum_{i=1}^{N} \sum_{c=1}^{C} (q_{i,c}^a - q_{i,c}^t)$, can be ignored since the probability outputs of both models sum to a constant value of $\mathbf{1}$. The second-order term, $\frac{1}{2N} \sum_{i=1}^{N} \sum_{c=1}^{C} \frac{(q_{i,c}^a - q_{i,c}^t)^2}{q_{i,c}^t}$, encourages the target model to maximize its output discrepancy with the adversary, especially on non-target classes where the target assigns lower probabilities. Since these smaller probabilities appear in the denominator, they induce larger regularization weights, thereby amplifying the effect of the loss on non-target classes. Although higher-order odd terms may introduce effects opposite to the desired behavior, they are typically suppressed by the preceding even terms, whose coefficients dominate.

In standard settings, the adversary model adopts the same architecture as the target model and typically produces a "single peak + uniform distribution" prediction pattern (Ma et al., 2022). This behavior arises from the use of one-hot labels in the cross-entropy loss, where the ground truth distribution is characterized by "single peak + uniform zero". The training behavior induced by NT can thus be summarized as follows: the cross-entropy loss drives the model to concentrate probability mass on the target class, while the nasty regularization term discourages it from replicating the adversary's uniform distribution across non-target classes. As a result, the target model redistributes the probability mass previously spread uniformly across all non-target classes onto a small subset of classes that are closer to the target (i.e., more compressible), ultimately leading to a sparse output probability distribution.

Our later experiments further show that the non-target class probabilities tend to be compressed onto semantically related classes (e.g., *cat* and *dog*), indicating that the model captures more generalizable semantics. In practice, even if the adversary's output deviates from the "single peak + uniform distribution" assumption, the differentiation-driven interaction between the adversary model and the target model still enforces a sparse probability distribution in the target model. However, the resulting peaks of the target model may not align with the adversary's multi-peak outputs, reflecting a different allocation of probability mass.

### 3.2 BENEFITS ON SPATIAL METRIC RELATIONSHIPS

The sparsity of output probabilities and the spatial metric behavior show obvious logical connection. Specifically, probability sparsity indicates that the model strongly favors the target class by assigning it substantially larger logits, while assigning much smaller logits to non-target classes. Let $i$ and $j$ denote the target class and a non-target class, respectively, with $(w_i, b_i)$ and $(w_j, b_j)$ representing the weights and bias of their corresponding linear classifiers. For a model exhibiting probability sparsity, we have:

$$w_i x + b_i \gg w_j x + b_j. \quad (9)$$

Here, $\gg$ rather than $>$ is used to emphasize the existence of a sufficiently large margin in a nasty-trained model, due to the saturation regions of activation functions such as *Sigmoid* and *Softmax*. This property relates to two key spatial metrics: (1) the distance from data points to the decision boundary, and (2) the minimum distance between classification boundaries. Both are measured by the magnitude of the output logits. Therefore, Equation 9 implies that robust models exhibit larger margins to the decision boundary and greater separation between hyperplanes, thereby achieving improved adversarial robustness. For clarity, we formalize the computation of these two distances in high-dimensional space.

**Distance from Data Points to the Decision Boundary.** To compute the distance from a data point to the decision boundary, we model the linear classification layer as a hyperplane in high-dimensional

space. The distance from a data point $x_i$ to the decision boundary of class $c$ is then given by the standard point-to-hyperplane distance formula:

$$\mathcal{D}_{data\_to\_bound} = \frac{|w_c \cdot x_i + b_c|}{||w_c||_2}. \tag{10}$$

*Analysis.* Intuitively, larger logits indicate greater distances from the decision boundary. Although this distance depends on the norm of the classification layer's weights ($||w_c||_2$), weight magnitudes are typically constrained by L2 regularization and thus vary within a limited range. Compared to the substantial shifts in logits caused by the saturation behavior of the *Softmax* function, variations in the model parameters are of much smaller magnitude. As a result, greater probability sparsity yields substantially larger distances between data points and their corresponding classification boundaries. In the experiment section, we empirically validate this hypothesis by quantifying the average distance of all samples within each class to the ten decision boundaries.

**Minimum Distance Between Classification Boundaries.** Furthermore, for the same data point $x_i$, the larger logit gap observed in the nasty-trained model can be attributed to greater discrepancy between classification parameters, defined as:

$$\mathcal{D}_{weight\_l2} = ||w_j - w_i||_2. \tag{11}$$

This increased weight discrepancy leads to larger inter-class distances in the classification layer. Specifically, the linear layer of deep model can be regarded as set of high-dimensional vectors. Although the exact distances between such vectors cannot be directly computed,the shortest distance between the corresponding class decision boundaries can be approximated using projection geometry. Specifically, the classification weight vectors for different categories can be treated as non-intersecting lines, and their shortest distance can be calculated as follows:

(1) Calculate the difference between the weight vectors: $\gamma = w_j - w_i$.

(2) Normalize the weight vectors to obtain unit directions: $d_i = \frac{w_i}{||w_i||_2}, d_j = \frac{w_j}{||w_j||_2}$.

(3) Estimate the shortest distance between the classification directions of classes $i$ and $j$ as:

$$\mathcal{D}_{shortest}^{i,j} = ||\gamma - (\gamma \cdot d_i) \cdot d_i||_2, \tag{12}$$

where $\gamma \cdot d_i$ is the scalar projection of $\gamma$ onto the direction $d_i$, and $(\gamma \cdot d_i) \cdot d_i$ is the corresponding projection vector.

Thus, the shortest distance between two classifier boundaries (e.g., $i$ and $j$) can be expressed as:

$$\mathcal{D}_{shortest}^{i,j} = ||w_i - w_j - (\frac{(w_j - w_i)}{||d_i||_2^2} \cdot d_i)||_2. \tag{13}$$

*Analysis.* Intuitively, larger inter-class weight gaps yield greater shortest distances between classification boundaries, making it more difficult for data points to shift between classes. As a result, adversaries must apply larger perturbations (i.e., higher attack budget) to induce a misclassification. To empirically validate this theoretical insight, we will later conduct a qualitative analysis in the experiments section, examining the relationship between inter-class weight differences and the distances between different classification boundaries.

## 4 EXPERIMENTS

### 4.1 EXPERIMENT SETUP

**Attacks Details.** Following common defense evaluation settings (Zhang et al., 2024b; Yue et al., 2024; Yin et al., 2024; Li et al., 2024; Zi et al., 2021; Zhou & Hua, 2024; Zhou et al.), we evaluate NAT against several widely used adversarial attacks: PGD with 10, 20, 50, and 100 steps (Madry et al., 2017), CW (Carlini & Wagner, 2017), and AutoAttack (Croce & Hein, 2020). For CIFAR10 and CIFAR100, the $L_\infty$ norm attack budget is set to $\epsilon = 8/255$, with perturbation step size $\eta_1 = 2/255$ and iterations $K = 10$. For ImageNet100, we set $\epsilon = 0.03$ and $\eta_1 = 2/255$.

**Training Details.** Following Ma et al. (2021; 2022), we choose a normally trained adversary model with the same architecture as the target model. The nasty regularization coefficient $\omega_a$ is set to

0.006, which is empirically validated as optimal in our ablation study. For the target model, we use SGD optimizer with momentum 0.9 and weight decay $5 \times 10^{-4}$. Training is conducted for 300 epochs with an initial learning rate of 0.1, decayed by a factor of 10 at epochs 160 and 240. Batch sizes are 512 for CIFAR datasets and 128 for ImageNet. All experiments are performed on NVIDIA GeForce RTX 4090 GPUs using PyTorch 1.12.1. The adversary model is set as the vanilla-trained counterpart of the target model under the basic setting, and we will discuss the model selection later. Furthermore, to ensure fair comparison with recent defenses that leverage diffusion-based adversarial data augmentation, we adopt the same augmentation strategy as in Wang et al. (2023), demonstrating the superiority of our NAT framework under identical conditions.

**Datasets and Backbones.** We evaluate NAT on two standard benchmark datasets, CIFAR-10 and CIFAR-100, which are widely used for adversarial robustness evaluation (Zhang et al., 2024b; Yue et al., 2024; Yin et al., 2024; Li et al., 2024; Zi et al., 2021; Zhou & Hua, 2024). To further validate its scalability, we also test NAT on a higher-resolution dataset, ImageNet100.

- CIFAR10 & CIFAR100. For both datasets, we employ ResNet-18 (He et al., 2016) and Wide-ResNet-34-10, which are commonly employed in adversarial defense evaluations.

- ImageNet100. For ImageNet100, we use ViT-Small (Alexey, 2020) as the backbone.

Our experimental comparisons primarily benchmark NAT against AGAIN (Jia et al., 2022a) and LAS-AT (Yin et al., 2023), which currently achieves the state-of-the-art performance (Zhang et al., 2024b; Yue et al., 2024; Yin et al., 2024; Li et al., 2024; Zi et al., 2021). All reported results correspond to the best outcomes over three independent runs.

Table 1: Test accuracy and robustness on CIFAR-10 and CIFAR-100 using Wide-ResNet-34-10.

| | CIFAR10 | | | | | | | CIFAR100 | | | | | | |
|---|---|---|---|---|---|---|---|---|---|---|---|---|---|---|
| | Clean | PGD-10 | PGD-20 | PGD-50 | C&W | AA | Avg. | Clean | PGD-10 | PGD-20 | PGD-50 | C&W | AA | Avg. |
| PGD-AT | 85.17 | 56.07 | 55.08 | 54.88 | 53.91 | 51.67 | 59.46 | 60.89 | 32.19 | 31.69 | 31.45 | 30.1 | 27.86 | 35.69 |
| TRADES | 85.72 | 56.75 | 56.10 | 55.90 | 53.87 | 53.40 | 60.28 | 58.61 | 29.20 | 28.66 | 28.56 | 27.05 | 25.94 | 33.00 |
| SAT | 87.97 | 50.31 | 49.86 | 48.79 | 48.65 | 47.48 | 55.51 | 62.82 | 28.1 | 27.17 | 26.76 | 27.32 | 24.57 | 32.79 |
| AWP | 85.57 | 58.92 | 58.13 | 57.92 | 56.03 | 53.90 | 61.74 | 60.38 | 34.13 | 33.86 | 33.65 | 31.12 | 28.86 | 37.0 |
| LBGAT | 88.22 | 56.25 | 54.66 | 54.30 | 54.29 | 52.23 | 59.99 | 60.64 | 35.13 | 34.75 | 34.62 | 30.65 | 29.33 | 37.52 |
| LAS-AWP | 87.74 | 61.39 | 60.16 | 59.79 | 58.22 | **55.52** | 58.80 | **64.89** | **37.11** | 36.36 | 36.13 | **33.92** | 30.77 | **39.86** |
| **NAT (best)** | **89.15** | **63.69** | **62.34** | **62.05** | **65.10** | 52.95 | **65.88** | 62.87 | 36.79 | **36.36** | **36.22** | 32.79 | **30.85** | 39.22 |
| | (89.10 ±0.13) | (63.43 ±0.21) | (62.24 ±0.17) | (61.77 ±0.25) | (64.89 ±0.30) | (52.48 ±0.41) | | (62.72 ±0.19) | (36.43 ±0.29) | (35.90 ±0.48) | (35.52 ±0.61) | (32.32 ±0.52) | (30.13 ±0.71) | |
| NAT (last) | 87.33 | 65.01 | 63.66 | 63.03 | 63.40 | 50.23 | 65.44 | 61.18 | 35.12 | 35.68 | 35.51 | 30.61 | 29.14 | 37.88 |
| | (87.15 ±0.14) | (64.86 ±0.21) | (63.42 ±0.16) | (62.79 ±0.27) | (63.04 ±0.29) | (49.87 ±0.45) | | (61.02 ±0.14) | (34.89 ±0.30) | (35.20 ±0.43) | (34.97 ±0.58) | (30.01 ±0.54) | (28.45 ±0.70) | |

## 4.2 MAIN RESULTS

We report the results for CIFAR10 and CIFAR100 on WRN-34-10 in Table 1, and on ResNet-18 in Table 2. The mean and standard deviation are reported in parentheses (mean±std). For comparative methods, the ResNet-18 results are taken from Yin et al. (2023), while the WRN-34-10 results are taken from Jia et al. (2022a). NAT is further evaluated on ImageNet100 (as presented in Appendix B), along with black-box attack settings (as presented in Appendix C). Results consistently show that NAT provides superior adversarial robustness, regardless of whether the backbone is a convolutional neural network or a ViT, and whether the dataset is low-resolution (CIFAR) or high-resolution (ImageNet). In the Appendix D, we further supplement the discussion on the effectiveness and superiority of NAT against adaptive attacks. Additionally, results of NAT with EDM-based data-augmentation are provide in the Appendix E. These results demonstrate that NAT is compatible with EDM-based augmentation, and can further exploit its potential to achieve even stronger robustness, compared with other defenses under the same augmentation conditions. Also, Figure 3 demonstrates the significant probability sparsity of Nasty VT & AT compared to Normal VT & AT.

## 4.3 ABLATION STUDY

We perform ablation studies from multiple perspectives to evaluate the convenience, efficiency, and effectiveness of NAT. All ablation experiments are conducted on CIFAR-10.

Table 2: Test accuracy and robustness on CIFAR-10 and CIFAR-100 dataset using ResNet-18.

| | CIFAR10 | | | | | | | | CIFAR100 | | | | | | | |
| --- | --- | --- | --- | --- | --- | --- | --- | --- | --- | --- | --- | --- | --- | --- | --- | --- |
| | Clean | PGD-10 | PGD-20 | PGD-50 | PGD-100 | C&W | AA | Avg. | Clean | PGD-10 | PGD-20 | PGD-50 | PGD-100 | C&W | AA | Avg. |
| PGD-AT | 84.25 | 46.88 | 46.56 | 44.85 | 44.76 | 45.75 | 41.69 | 50.67 | 62.34 | 21.24 | 21.38 | 21.05 | 21.01 | 22.15 | 19.76 | 26.99 |
| MART | 81.61 | 52.38 | 51.28 | 50.93 | 50.80 | 47.77 | 46.09 | 54.40 | 55.14 | 28.52 | 28.08 | 27.79 | 27.91 | 25.65 | 24.04 | 31.01 |
| TRADES | 83.64 | 52.05 | 50.67 | 50.38 | 50.20 | 49.68 | 48.41 | 55.00 | 58.18 | 28.71 | 28.25 | 28.10 | 27.99 | 24.22 | 24.03 | 31.35 |
| FAT | 87.32 | 45.80 | 43.53 | 43.11 | 42.98 | 43.50 | 40.76 | 49.57 | 61.61 | 19.33 | 18.35 | 18.08 | 17.98 | 19.31 | 17.38 | 24.57 |
| LBGAT | 85.73 | 53.12 | 52.05 | 51.78 | 51.68 | 50.63 | 49.04 | 56.29 | 56.78 | 32.84 | 32.21 | 32.11 | 32.07 | 27.46 | 26.39 | 34.26 |
| CAS | 86.24 | 51.38 | 51.49 | 51.77 | 51.04 | 53.66 | 46.69 | 56.03 | 64.04 | 31.66 | 31.55 | 31.26 | 31.02 | 34.82 | 24.40 | 35.53 |
| AWP | 79.45 | 55.04 | 54.47 | 54.36 | 54.30 | 51.17 | 49.40 | 56.88 | 54.00 | 31.78 | 31.49 | 31.44 | 31.74 | 28.20 | 26.19 | 33.54 |
| LAS-AT | 82.39 | 54.74 | 53.70 | 53.70 | 53.72 | 51.96 | 49.94 | 57.16 | 58.38 | 32.32 | 31.89 | 31.82 | 31.77 | 28.48 | 26.84 | 34.5 |
| AGAIN-AWP | 86.52 | 59.99 | 59.35 | 59.11 | 58.85 | 61.19 | **51.89** | 62.41 | **64.51** | 35.58 | 35.44 | 35.39 | 35.08 | **40.02** | 28.69 | 39.24 |
| **NAT (best)** | **90.86** (90.77 ±0.15) | **62.37** (62.12 ±0.22) | **60.94** (60.81 ±0.27) | **60.19** (59.91 ±0.19) | **59.91** (59.72 ±0.32) | **62.54** (62.24 ±0.34) | 50.18 (50.00 ±0.46) | **63.85** | 64.02 (63.77 ±0.22) | **36.82** (36.69 ±0.22) | **36.63** (36.39 ±0.33) | **36.44** (36.27 ±0.41) | 35.16 (34.84 ±0.56) | 37.98 (37.77 ±0.51) | **28.82** (28.54 ±0.73) | **39.26** |
| NAT (last) | 90.28 (90.07 ±0.18) | 61.86 (61.72 ±0.22) | 60.00 (59.72 ±0.21) | 59.44 (59.26 ±0.36) | 58.89 (58.57 ±0.29) | 61.67 (61.46 ±0.41) | 48.96 (48.65 ±0.51) | 63.01 | 61.80 (61.63 ±0.23) | 34.55 (34.30 ±0.33) | 34.43 (34.29 ±0.41) | 34.19 (33.92 ±0.39) | 34.07 (33.91 ±0.57) | 36.71 (36.42 ±0.48) | 27.17 (26.97 ±0.68) | 37.56 |

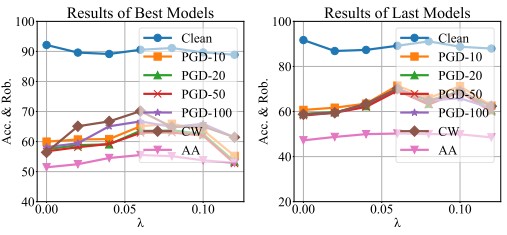

Figure 2: Ablation study of $\lambda$ on CIFAR10 dataset.

**Selection of the Hyper-Parameter $\lambda$.** In the original NT, the default value of $\lambda$ is set to 0.08. However, this choice may not be optimal under adversarial training. To investigate its effect, we vary $\lambda$ in the range [0, 0.12] with a step size of 0.02 and report the results in Figure 2. The results indicate that for both the best and last models, accuracy and robustness first increase and then decrease, reaching the peak performance at $\lambda = 0.06$. Importantly, across all tested values, introducing nasty regularization consistently improves robustness, as shown from the markedly poorer performance observed at $\lambda = 0$.

**Impact of Adversary Model Structure.** In the original setup, the adversary model is configured as the naturally trained counterpart of the target model. We further investigate the effect of varying adversary model architectures on NAT in Appendix F. Overall, we find that adversary models with different structures consistently contribute to robustness improvements, offering flexible options for NAT. Please refer to Appendix F for detailed discussion.

**Impact of Adversary Model State.** We analyze the impact of different adversary model parameter states on NAT in Appendix G, including random initialization, vanilla training, adversarial training, and SAM regularization. Overall, while all parameter states contribute to robustness improvements, some do not exhibit the characteristic "single peak + uniform distribution" probability pattern. Detailed discussions are provided in Appendix G.

## 4.4 Verification of Spatial Metric Relationships.

We quantitatively validate the conclusions presented in the in-depth analysis section, including (1) the increased distances from data points to classification boundaries, (2) the enlarged inter-class weight gaps, and (3) the greater shortest distances between classification hyperplanes. All quantification results are achieved on the test set.

**Distance from Data Point to Classification Boundary.** We measure the distance of all samples from class **0** to each classification boundaries, as shown in Figure 6. To better illustrate the metric relationship between data points and classification boundaries, we remove the absolute value operation in Equation 10. Intuitively, the correctly classified target class should exhibit a larger positive distance, whereas non-target classes should yield smaller or even negative distances. The robust model indeed shows greater distances to the decision boundaries compared to standard models. Moreover, the robust model tends to assign positive logits (i.e., positive distances) to semantically related classes (e.g., *dog* and *cat*), while dissimilar categories (e.g., *automobile* and *ship*) consistently yield negative distances. This behavior indicates that NAT does not simply overfit the training data but tends to capture invariant semantic structures shared across similar categories while preserving higher inter-class separability.

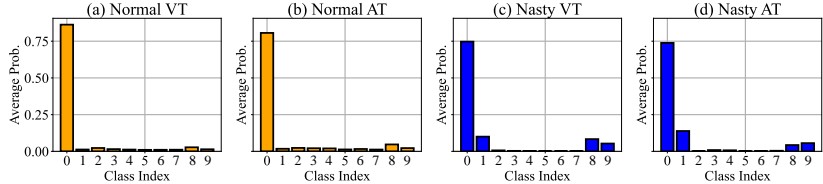

Figure 3: Verification of sparsity of model output probability.

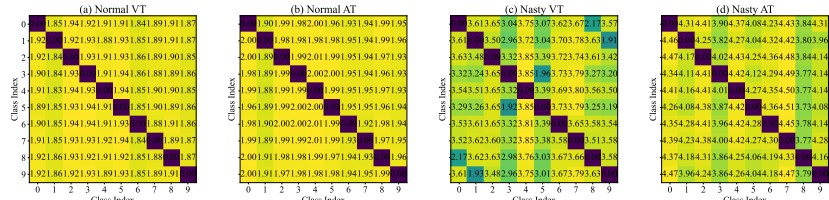

Figure 4: Symmetric matrix of the shortest distance among each boundary.

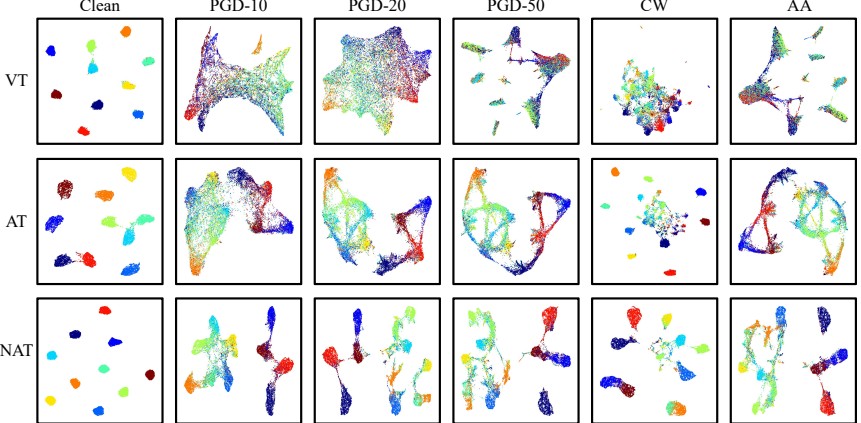

Figure 5: UMAP visualization of feature distributions for Vanilla Trained model (VT), Adversarial Trained model(AT) and Nasty Adversarial Trained model (NAT).

**Inter-Class Gap in the Weights of the Classification Layer.** We use the L2 norm to quantify the magnitude of inter-class weight differences, ignoring directional effects. As shown in Figure 7, the results indicate that the nasty model produces substantially larger inter-class weight separations in the classification layer compared to the normal model.

**Shortest Distance Among Linear Layers.** Following the in-depth analysis, we compute the shortest inter-class distances in the classification layers of both natural and robust models under normal and nasty settings. As shown in the symmetric matrices of Figure 4, NAT consistently exhibits substantially larger shortest inter-class distances compared to standard models.

These findings validate our hypothesis regarding the spatial metric preference: adversaries must expend a larger attack budget to force misclassification, thereby enhancing robustness. In addition, a visualization of feature distributions using UMAP, provided in Figure 5, further confirms NAT's spatial preference, characterized by larger inter-class separation and smaller intra-class compactness.

## 5 FURTHER DISCUSSION

We further examine two additional questions: (a) Does employing an ensemble model as the adversary model provide further gains for NAT? (b) Can NAT be extended to robustness distillation, and if so, does it achieve similar improvements or hinder the student model's learning process as in the

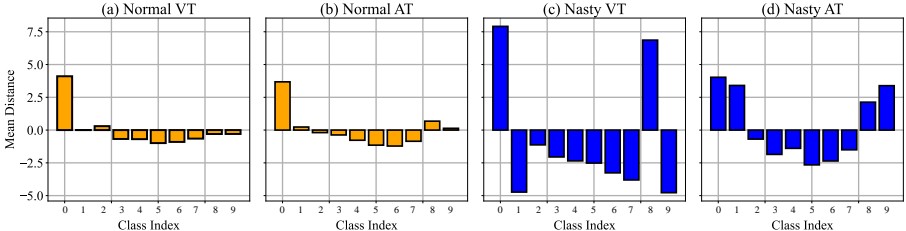

Figure 6: Average distance of class 0 samples in CIFAR10 dataset to all classification boundaries.

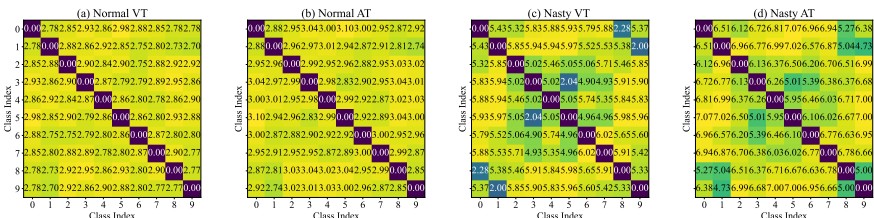

Figure 7: Symmetric matrix of L2 distances between classification layer weights.

nasty training? Our findings suggest that while the ensemble adversary model provides marginal improvements, it is not necessary. Moreover, NAT can provide a degree of undistillable protection for robust teachers, though further investigation is needed to strengthen this property. We also compare NAT against explicit regularizers such as entropy regularization and logits norm regularization, highlighting its irreplaceability. In addition, we provide a brief discussion of NAT's training cost, showing that it introduces only minimal overhead. Finally, we outline the limitations of NAT. More detailed analyses of these issues are presented in Appendix H,I, J, K, M.

## 6 CONCLUSION

In this paper, we propose nasty adversarial training (NAT), a simple yet effective regularizer for adversarial training that leverages the probabilistic sparsity prior. We first provide a theoretical analysis of how NAT induces probabilistic sparsity and examine its role in enhancing robustness from a metric perspective. Extensive experiments demonstrate that NAT achieves state-of-the-art adversarial robustness. Additional analyses further highlight its simplicity, efficiency, and irreplaceability.

## ETHICS STATEMENT

Our research on NAT addresses an open problem in AI using public datasets. We declare no potential conflicts of interest, and the work raises no issues related to bias, fairness, privacy, security, or legal compliance. The study, which is purely methodological and requires no IRB approval, is conducted with the sole intention of improving model robustness against adversarial attacks and presents no malicious ethical threats.

## ACKNOWLEDGMENTS

This work was supported in part by the National Natural Science Foundation of China under Grants 62572150, 62372137, and 62472117, the Guangdong Basic and Applied Basic Research Foundation under Grant 2024A1515012299, the Major Key Project of PCL under Grant PCL2024A05, and the Shenzhen Science and Technology Program under Grant JCYJ20230807094411024.

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

# APPENDIX

## A    RELATED WORKS

### A.1    ADVERSARIAL ATTACKS

Adversarial attacks aim to introduce imperceptible perturbations to input data, causing a well-trained model to produce incorrect predictions. Depending on the amount of information accessible to the attacker, adversarial attacks are generally categorized into white-box and black-box settings. In white-box attacks, the attackers has fully access to the target model. Common methods include gradient-based approaches (Goodfellow et al., 2014; Dong et al., 2018; Madry et al., 2017), classifier-based methods (Moosavi-Dezfooli et al., 2016), and optimization-based techniques (Carlini & Wagner, 2017). In contrast, black-box attacks assume limited prior knowledge of the target model and are typically classified into score-based (Chen et al., 2017), decision-based (Brendel et al., 2017), and transfer-based attacks (Xie et al., 2019b; Zou et al., 2020). Among them, transfer-based attacks involve training a surrogate model to simulate the target model's behavior and are widely used to evaluate the black-box adversarial robustness of DNNs.

### A.2    ADVERSARIAL DEFENSES

Adversarial defenses aim to improve a model's accuracy on adversarial examples. AT (Goodfellow et al., 2014; Zhang et al., 2019; Dong et al., 2018; Madry et al., 2017; Jia et al., 2022a; Wang et al., 2019) is widely regarded as one of the most effective defense. RD (Goldblum et al., 2020; Zhu et al., 2021; Zi et al., 2021) is later proposed to transfer robustness from a large, robust model to a smaller, more efficient student model. Recently, AT are further explored, such as fairness issue (Zhang et al., 2024b), attention maps (Yin et al., 2023), and learnable strategies (Jia et al., 2022a). Data-free adversarial defense (Wang et al., 2024; Zhou et al.; Lee et al., 2024) are explored to achieve adversarial robustness in scenarios with limited data. Additionally, internal maximization methods like fast adversarial training (FAT) (Jia et al., 2022b; Huang et al., 2023) have been developed to enhance the efficiency of AT without significantly sacrificing robustness. In this paper, we focus on AT and propose a simple yet effective regularization strategy from the perspective of output probability sparsity, which achieves state-of-the-art adversarial robustness with little additional computational overhead.

Table 3: The results of several advanced adversarial defense using the EDM-based data augmentation.

| Dataset | Architecture | Method | Generated | Batch | Epoch† | Clean | AA |
|---|---|---|---|---|---|---|---|
| CIFAR-10 ($l_\infty, \epsilon = 8/255$) | WRN-34-20 | (Rice et al., 2020) | ✗ | 128 | 200 | 85.34 | 53.42 |
| | WRN-34-10 | (Zhang et al., 2020a) | ✗ | 128 | 120 | 84.52 | 53.51 |
| | WRN-34-20 | (Pang et al., 2020) | ✗ | 128 | 110 | 86.43 | 54.39 |
| | WRN-34-10 | (Wu et al., 2020) | ✗ | 128 | 200 | 85.36 | 56.17 |
| | WRN-70-16 | (Gowal et al., 2020) | ✗ | 512 | 200 | 85.29 | 57.14 |
| | WRN-34-10 | (Sehwag et al., 2021) | 10M | 128 | 200 | 87.00 | 60.60 |
| | WRN-28-10 | (Rebuffi et al., 2021) | 1M | 1024 | 800 | 87.33 | 60.73 |
| | | (Pang et al., 2022) | 1M | 512 | 400 | 88.10 | 61.51 |
| | | (Gowal et al., 2021) | 100M | 1024 | 2000 | 87.50 | 63.38 |
| | | (Wang et al., 2023) | 1M | 512 | 400 | 91.12 | 63.35 |
| | | | 1M | 1024 | 800 | 91.43 | 63.96 |
| | | | 50M | 2048 | 1600 | 92.27 | 67.17 |
| | | | 20M | 2048 | 2400 | 92.44 | 67.31 |
| | WRN-70-16 | (Pang et al., 2022) | 1M | 512 | 400 | 88.57 | 63.74 |
| | | (Rebuffi et al., 2021) | 1M | 1024 | 800 | 88.54 | 64.20 |
| | | (Gowal et al., 2021) | 100M | 1024 | 2000 | 88.74 | 66.11 |
| | | (Wang et al., 2023) | 1M | 512 | 400 | 91.98 | 65.54 |
| | | | 5M | 512 | 800 | 92.58 | 67.92 |
| | | | 50M | 1024 | 2000 | 93.25 | 70.69 |
| | WRN-34-10 | (Wu et al., 2020) | ✗ | 128 | 200 | 60.38 | 28.86 |
| | WRN-70-16 | (Gowal et al., 2020) | ✗ | 512 | 200 | 60.86 | 30.03 |
| | WRN-34-10 | (Sehwag et al., 2021) | 1M | 128 | 200 | 65.90 | 31.20 |
| | WRN-34-10 | () | ✗ | 128 | 200 | 0 | 0 |
| | | | AutoAug | 128 | 200 | 0 | 0 |
| | WRN-34-10 | **Ours** | ✗ | 512 | 300 | 89.15 | 52.95 |
| | | | 1M | 1024 | 600 | 90.44 | 55,41 |
| | | | 5M | 1024 | 600 | 91.01 | 57.24 |
| CIFAR-100 ($l_\infty, \epsilon = 8/255$) | WRN-28-10 | (Pang et al., 2022) | 1M | 512 | 400 | 62.08 | 31.40 |
| | | (Rebuffi et al., 2021) | 1M | 1024 | 800 | 62.41 | 32.06 |
| | | (Wang et al., 2023) | 1M | 512 | 400 | 68.06 | 35.65 |
| | | | 50M | 2048 | 1600 | 72.58 | 38.83 |
| | WRN-70-16 | (Pang et al., 2022) | 1M | 512 | 400 | 63.99 | 33.65 |
| | | (Rebuffi et al., 2021) | 1M | 1024 | 800 | 63.56 | 34.64 |
| | | (Wang et al., 2023) | 1M | 512 | 400 | 70.21 | 38.69 |
| | | | 50M | 1024 | 2000 | 75.22 | 42.67 |
| | WRN-34-10 | () | ✗ | 128 | 200 | 0 | 0 |
| | | | AutoAug | 128 | 200 | 0 | 0 |
| | WRN-34-10 | **Ours** | ✗ | 512 | 300 | 62.87 | 30.85 |
| | | | 1M | 1024 | 800 | 64.74 | 33.08 |
| | | | 5M | 1024 | 800 | 66.63 | 36.71 |

## B    RESULTS ON IMAGENET-100

In this section, we report the robustness performance of NAT on ImageNet100. Due to the lack of direct comparisons, we primarily compare NAT with natural models and adversarial training models. The experimental results are shown in Table 4. Under the same experimental setup, compared to smaller datasets like CIFAR10, NAT demonstrates significant robustness increments on ImageNet100.

## C    RESULTS ON BLACK-BOX ATTACKS

In this section, we provide a simple evaluation of NAT's adversarial robustness against black-box attacks based on CIFAR10. We mainly follow the black-box attack settings in RSLAD, which test both the transfer attack and query-based attack. For transfer attack, we generate the adversarial examples by PGD-20 and CW on an adversarially pre-trained surrogate ResNet-50. The maximum perturbation is also set to 8/255. For query-based attack, we use the strong Square attack. The

Table 4: Test accuracy and robustness of the ImageNet-100 dataset on ViT-small.

|  | Clean | PGD-10 | PGD-20 | PGD-50 | PGD-100 | C&W | AA | Ave. |
|---|---|---|---|---|---|---|---|---|
| Vanilla Training | **96.23** | 8.23 | 6.54 | 4.58 | 4.14 | 7.97 | 1.68 | 18.48 |
| PGD-AT | 95.69 | 64.44 | 62.32 | 39.80 | 39.71 | 41.64 | 37.12 | 54.38 |
| NAT (best) | 94.92 | **67.07** | **66.84** | **43.22** | 44.16 | **49.82** | **45.44** | **58.78** |
| NAT (last) | 94.87 | 64.77 | 66.37 | 42.13 | **45.07** | 48.53 | 44.40 | 58.01 |

Table 5: Black-box robustness on CIFAR10.

| Methods | ResNet-18 | | | MobileNetV2 | | |
|---|---|---|---|---|---|---|
|  | PGD$_S$ | CW | Square | PGD$_S$ | CW | Square |
| SAT | 60.84 | 60.52 | 54.27 | 60.46 | 59.83 | 53.94 |
| ARD | 63.49 | 63.05 | 56.89 | 62.13 | 61.85 | 55.60 |
| IAD | 62.78 | 62.26 | 56.62 | 61.57 | 61.25 | 55.45 |
| Trades | 62.20 | 61.75 | 55.13 | 60.90 | 60.23 | 53.46 |
| RSLAD | 64.11 | 63.84 | 57.90 | 63.30 | 63.20 | 56.70 |
| LAS-AT | 66.42 | 65.41 | 60.21 | 66.14 | 65.42 | 58.87 |
| **NAT** | **67.19** | **66.77** | **60.23** | **67.82** | **66.49** | **59.21** |

Table 6: Evaluation on the adaptive attacks.

| Backbones | Defenses | CIFAR10 | | | | | CIFAR100 | | | | |
|---|---|---|---|---|---|---|---|---|---|---|---|
|  |  | APGD-10 | APGD-20 | APGD-50 | APGD-100 | Ave. | APGD-10 | APGD-20 | APGD-50 | APGD-100 | Ave. |
| WRN3410 | VT | 12.17 | 9.88 | 9.02 | 8.88 | 9.98 | 2.91 | 2.68 | 2.63 | 2.51 | 2.68 |
|  | AT-PGD | 57.57 | 54.52 | 53.36 | 53.14 | 54.64 | 28.21 | 28.00 | 27.95 | 27.80 | 27.99 |
|  | **NAT** | **61.48** | **59.02** | **58.27** | **58.11** | **59.22** | **33.14** | **32.41** | **32.01** | **31.77** | **32.33** |
| ResNet18 | VT | 10.37 | 9.17 | 8.77 | 8.71 | 9.25 | 4.47 | 4.09 | 3.95 | 3.90 | 4.10 |
|  | AT-PGD | 53.77 | 52.07 | 51.60 | 51.48 | 52.23 | 27.33 | 26.83 | 26.67 | 26.65 | 26.87 |
|  | **NAT** | **61.00** | **58.25** | **58.11** | **57.75** | **58.77** | **33.62** | **32.72** | **32.14** | **32.01** | **32.62** |

experimental results are shown in Table 5. The results in the table include both adversarial training methods and robustness distillation methods. To maintain consistency, we performed NAT training on MobileNetV2 for evaluation. It can be observed that our proposed NAT significantly outperforms the provided mainstream adversarial training methods and robustness distillation methods in black-box attacks, showing its superiority.

# D  EVALUATION ON THE ADAPTIVE ATTACKS

In this section, we discuss the effectiveness of NAT against adaptive attacks. This discussion is based on the assumption that the attacker is fully aware of the defender's NAT defense paradigm and, consequently, uses the objective loss function of NAT as the target function for their attacks. The objective function of an adaptive attacker can be formalized as:

$$\arg \max_{\|\delta\|_p \leqslant \epsilon} \left[ XE \left( \sigma \left( p_{f_{\theta_t}}(x + \delta) \right), y \right) - \omega_a KL \left( \sigma_{\tau_a} \left( p_{f_{\theta_t}}(x + \delta) \right), \sigma_{\tau_a} \left( p_{f_{\theta_a}}(x + \delta) \right) \right) \right] \quad (14)$$

Such attack strategy is constructed using the PGD framework, with its hyperparameters remaining consistent with those specified in the main body of the paper. We similarly evaluate the effectiveness of NAT on both ResNet-18 and WideResNet-34-10 architectures, utilizing the CIFAR-10 and CIFAR100 datasets, and compare its performance against both Vanilla Training and PGD-AT. The experimental results are summarized in Table 6. NAT consistently demonstrates effective robustness against adaptive attacks, significantly outperforming PGD-AT. This robust performance underscores the reliability of NAT's defensive capabilities. In essence, unlike early defenses that rely on randomization, NAT remains fundamentally rooted in the adversarial training paradigm, which inherently avoids the pitfalls associated with "obfuscated gradients".

# E   COMBINING NAT WITH EDM-BASED DATA AUGMENTATION

In this section, we provide the experimental results of NAT using EDM-based data augmentation (Wang et al., 2023), with comparisons to state-of-the-art (SOTA) adversarial defenses under the same data augmentation strategy. The results are shown in Table 3. According to the conclusion of Wang et al. (2023), under conditions of large-scale data augmentation, a larger batch size and more training epochs can significantly improve model performance (Wang et al., 2023). Still, with identical data augmentation conditions (i.e., generated data number, batch size, and epoch number), NAT can achieve optimal adversarial robustness performance. This demonstrates the inherent superiority of NAT, confirms its compatibility with such augmentation, and reveals how they mutually explore each other's potential for enhancing adversarial robustness.

Table 7: The impact of different adversary model architecture. The target backbone is WRN-34-10.

|  |  | Clean | PGD-10 | PGD-20 | PGD-50 | PGD-100 | C&W | AA | Ave. |
|---|---|---|---|---|---|---|---|---|---|
| Adversary Model Structure | WRN-34-10 | 89.15 | 63.69 | 62.34 | 62.05 | 62.01 | 65.10 | 52.95 | 65.32 |
|  | DN121 | 87.95 | 66.00 | 64.39 | 63.62 | 63.42 | 65.46 | 48.47 | 65.61 |
|  | RN18 | 88.09 | 66.73 | 64.88 | 64.09 | 63.00 | 67.36 | 48.43 | 66.08 |
|  | RN50 | 88.12 | 68.04 | 66.71 | 66.15 | 65.08 | 67.47 | 46.67 | 66.89 |
| Ensemble Models |  | 88.94 | 69.12 | 67.34 | 67.19 | 66.37 | 67.84 | 46.00 | 67.64 |

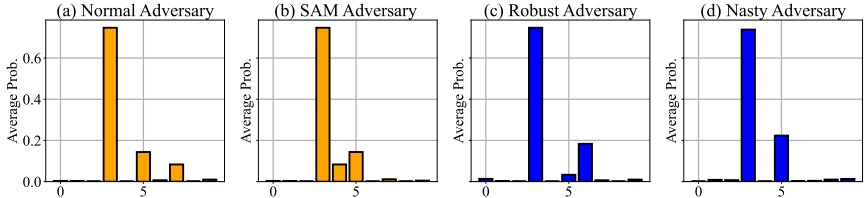

Figure 8: For the NAT with different adversary model structures, the output probability of "cat" class.

Table 8: The impact of different adversary model parameter state. The target backbone is WRN-34-10.

|  |  | Clean | PGD-10 | PGD-20 | PGD-50 | C&W | AA | Ave. |
|---|---|---|---|---|---|---|---|---|
| Para. State | VT | 89.15 | 63.69 | 62.34 | 62.05 | 65.10 | 52.95 | 65.88 |
|  | SAM | 84.44 | 67.63 | 63.71 | 63.39 | 65.01 | 47.69 | 65.31 |
|  | AT | 85.90 | 67.66 | 65.96 | 64.92 | 63.83 | 53.42 | 66.94 |
|  | Nasty | 82.94 | 68.16 | 64.14 | 64.02 | 63.04 | 49.53 | 65.30 |

Table 9: Training time statistics for one epoch.

|  | CIFAR10 | | CIFAR100 | |
|---|---|---|---|---|
|  | ResNet-18 | WRN-34-10 | ResNet-18 | WRN-34-10 |
| Vanilla Training | ≈ 15.3 s | ≈ 90 .2s | ≈ 15.7s | ≈ 101.3s |
| Adversarial Training | ≈ 108.6s | ≈ 1047.2 s | ≈ 110.9s | ≈ 1197.2s |
| MID (Zhang et al., 2024a) | ≈ 2530±10s | - | ≈ 3460±20s | - |
| **NAT** | ≈ 129.8 s | ≈ 1165.8 s | ≈ 127.3s | ≈ 1328.2s |

# F   IMPACT OF ADVERSARY MODEL STRUCTURE

In the original setup of NT, the adversary model is configured as a vanilla-trained model with the same structure as the target model. Here, we discuss the impact of adversary models with different

Table 10: The impact of NT to robustness distillation.

|  |  | Clean | PGD-10 | PGD-20 | PGD-50 | PGD-100 | C&W | AA | Ave. |
|---|---|---|---|---|---|---|---|---|---|
| RSLAD | From AT Teacher | 89.23 | 50.57 | 50.19 | 50.09 | 48.86 | 47.15 | 45.77 | 54.55 |
|  | From NAT Teacher | 87.97 | 47.50 | 47.24 | 47.20 | 47.15 | 48.86 | 42.64 | 52.65 |
| ARD | From AT Teacher | 79.43 | 38.84 | 38.21 | 39.93 | 39.87 | 36.61 | 34.12 | 43.85 |
|  | From NAT Teacher | 77.67 | 37.66 | 37.52 | 37.46 | 37.45 | 38.78 | 30.43 | 42.42 |
| IAD | From AT Teacher | 74.72 | 42.01 | 41.91 | 41.89 | 41.82 | 44.41 | 37.55 | 46.33 |
|  | From NAT Teacher | 74.29 | 41.64 | 41.61 | 41.58 | 41.48 | 43.39 | 36.25 | 45.74 |

Table 11: Comparing NAT with explicit regularizers.

| Methods | Clean | PGD10 | PGD20 | PGD50 | PGD-100 | C&W | AA | Avg. |
|---|---|---|---|---|---|---|---|---|
| VT | 95.03 | 11.72 | 10.52 | 10.13 | 10.09 | 11.42 | 7.81 | 22.38 |
| NT | 94.37 | 16.57 | 14.65 | 13.38 | 12.92 | 13.04 | 8.84 | 24.82 |
| AT | 84.25 | 46.88 | 46.56 | 44.85 | 44.76 | 45.75 | 41.69 | 50.67 |
| AT+LASSO | 88.06 | 41.45 | 39.77 | 39.08 | 38.78 | 40.18 | 30.54 | 45.40 |
| AT+EMR | 90.4 | 55.63 | 54.67 | 54.32 | 54.22 | 53.04 | 47.77 | 58.57 |
| AT+NNR | 92.13 | 53.83 | 52.59 | 52.1 | 51.97 | 52.04 | 46.53 | 57.31 |
| AT+mixup | 82.5 | 47.66 | 46.88 | 46.57 | 46.55 | 48.49 | 38.07 | 50.95 |
| NAT | 90.86 | 62.37 | 60.94 | 60.19 | 59.91 | 62.54 | 50.18 | 63.85 |

structures on the robustness increments. We select WRN-34-10 as the backbone of the target model for ablation analysis and use DenseNet-121, ResNet-18, and ResNet-50 as the backbones of the adversary model, in order to analyze the effects of different structures and different capacities. For ease of comparison, we also report results for the adversary model that is isomorphic to the target model. The experimental results, as shown in Table 7, indicate that various structures of adversary models, whether isomorphic or heterogeneous, can provide a stable robustness increment to the target model. Moreover, adversary models with larger capacities seem to demonstrate better robustness increments for the target model. For instance, ResNet50, serving as the nasty model, can provide better robustness increments compared to ResNet-18 to the target model.

## G  IMPACT OF ADVERSARY MODEL STATE

In the original setup, the adversary model is configured as a natural model with the same structure as the target model. Here, we discuss the impact of different model parameter states on robustness improvement, which includes random initialization, vanilla training, adversarial training, SAM regularization, and Nasty model. The experimental results are shown in Table 8, demonstrating that different model parameters can all provide robustness improvements for NAT.

An interesting question is that the nasty model no longer shows the "single peak + uniform distribution" probability distribution, yet still provides robustness improvement. We speculate that the output probabilities of the nasty model exhibit "a few peaks & uniform zero distribution", thus allowing the target model to achieve a sparse probability distribution by moving away from the uniform zero distribution. However, the non-target class peaks of the target model might have different labels compared to the non-target class peaks when using a regular model as the adversary.

We validated this hypothesis in Figure 8. For the "cat" class, when a normal model is used as the adversary model, the suboptimal peaks are "dog" and "horse." When the nasty model is used as the adversary model, the suboptimal peak concentrates on "dog". Similarly, suboptimal peak variations also appear when SAM models and Robust models are used as adversary models. However, All the setups can achieve a sparse probability distribution and robustness improvement on the target model. Such results confirm our hypothesis: different adversary models may lead the model to learn different suboptimal classes, but they consistently cause the target model to learn a sparse probability distribution. The discussion above may also partially explains why feeding the adversary model with both natural examples and adversarial examples can provide a significant adversarial robustness

improvement for the target model, even though the outputs of the adversary model on adversarial examples may not satisfy the "single peak" assumption.

## H   ANALYSIS OF TIME COSTS

In this subsection, we briefly discuss the computational cost of NAT. In recent studies on adversarial robustness, MID (Zhang et al., 2024a) provides an intuitive quantification of computational time cost by measuring the time required to train one epoch. Following the setup in MID, we also present a simple quantification of the computational time cost, as shown in Table 9. Intuitively, NAT introduces only limited additional computation compared to standard adversarial training, primarily attributed to the inference time of the adversary model. We further demonstrate that the architectural design of the adversary model has negligible impact on the target model. It is worth mentioning that the time cost reported in Table 9 does not include the time cost of performing Vanilla Training (VT) on the adversary model. This omission is primarily based on the following considerations: (a) The structure and training settings of the adversary model are flexible and can generally bring performance gains to the model (as shown in Table 7 and Table 8). (b) Compared to adversarial training, the time cost of vanilla training is negligible. In cases where the target model and the adversary model share the same architecture, the time cost reported for "vanilla training" in the first row can serve as a reference for the VT time cost of the adversary model. In conclusion, we argue that the extra computational cost of NAT remains controllable and acceptable compared to standard adversarial training.

## I   DOES ENSEMBLE ADVERSARY MODEL HELP?

In machine learning, a natural approach to model enhancement is ensemble learning. In this section, we discuss the robustness performance of NAT when the adversary model is an ensemble model. Specifically, we choose ResNet-18, DenseNet121, and VGG11 to form the ensemble model. We use the same ensemble method as A, guiding the nasty regularization with the mean of the logits output by the three models. The experimental results are shown in the last line of Table 7.

Compared to the standard setup, the ensemble model shows a slight incremental advantage. This is an expected result, as models of any structure tend to exhibit a "single peak + uniform distribution" probability output preference. Averaging the logits of different models further smooths the uniform distribution. We have analyzed that moving away from this uniform distribution can bring robustness improvements to NAT. However, given the significantly increased computational cost, this slight increment may suggest that an ensemble adversary model is not necessary.

## J   TRANSFER NAT TO ROBUSTNESS DISTILLATION

In this section, we discuss an open question: Can NAT be transferred to robust distillation to achieve incremental improvements? Or does it, like a regular nasty model, prevent the robust model from being distilled by student models?

As we mentioned, the original purpose of NT is to make the nasty model difficult for student models to learn simply. However, whether this property can be transferred to NAT is still unclear. If this property is transferable, NAT could also serve as a method to protect the copyright of robust models, preventing robust teachers from being distilled into student models without authorization. We briefly discuss this issue in this section.

Specifically, we use WRN-34-10 trained with standard AT and NAT as the robust teacher models, and randomly initialized ResNet-18 as the student model. We apply RSLAD, ARD, and IAD as robustness distillation methods to perform robustness distillation on the student model. The experimental results are shown in Table 10. As a teacher model, although the NAT teacher shows better robustness performance than the AT teacher, its student model's robustness performance is relatively poor. This indicates that NAT has a certain tendency to avoid being learned, which can also serve as a method to protect the intellectual property of robust teachers. However, this protective effect is still far from the ideal of an unlearnable teacher, as the student model still acquires a considerable level

of robustness performance. There remains significant space for research in the intellectual property protection of adversarial robust models.

## K    COMPARING NAT WITH EXPLICIT REGULARIZERS

NAT can be considered as integrating probability sparsity regularization into AT. In this section, we explain the differences between NAT and several other explicit regularization methods, as well as the advantages of NAT.

To begin with, we need to clarify the distinction between the proposed probability sparsity and model sparsity. From the conceptual perspective, model sparsity generally refers to the sparsity of model parameters, which reduces the redundancy of the model, while probability sparsity means that the probabilities of irrelevant categories, calculated based on the model's output logits, approach zero. From the effect perspective, model sparsity aims to decrease model complexity and reduce redundant features, thereby enhancing generalization. The effect of probability sparsity is more intuitive: it induces the target model to output probabilities only for the target category or similar categories, while reducing confidence in irrelevant categories. This allows the model to learn more generalized and semantically coherent feature representations and classification preferences. Thus, there is an essential difference between the two. A common regularization method for model sparsity is LASSO regularization, which can be combined with adversarial training and formalized as:

$$\mathcal{L} = \mathcal{L}_{\text{AT}} + \lambda_{\text{LASSO}} \cdot \sum_{w \in \theta} |w| \tag{15}$$

where $\lambda_{\text{LASSO}}$ is set as 0.0001. We compare the experimental results of NAT and Lasso regularization in Table 11, where NAT still shows better results. This proves that LASSO regularization cannot achieve the probability sparsity that NAT can provide, making it difficult to attain similar robustness increments.

Furthermore, in order to demonstrate the irreplaceability of NAT, we explain the differences between NAT and directly constraining the probability outputs. One intuitively potential method to achieve probability sparsity is to directly add explicit regularization at the output end, such as minimizing the output entropy:

$$\mathcal{L} = \mathcal{L}_{\text{AT}} + \lambda_{\text{EMR}} \cdot \sum_{i=1}^{N} p_i \log p_i \tag{16}$$

which we call entropy minimization regularization(EMR) later. Also, one can regularize negative of the L2 norm of the output values as:

$$\mathcal{L} = \mathcal{L}_{\text{AT}} - \lambda_{\text{NNR}} \cdot \sqrt{\sum_{i=1}^{C} p_i^2} \tag{17}$$

which we call negative norm regularization(NNR) later.

Intuitively, both EMR and NNR induce the model to output sharp probability distributions and achieve optimal solutions when the model outputs one-hot probabilities. Although they also achieve probability sparsity in form, this sparsity is suboptimal. The overly rigid one-hot constraint resembles overfitting. EMR and NNR amplify the cross-entropy loss's "winner takes all" preference, causing the model to be overly confident in the current training data and label, which results in a loss of generalization capability. At the same time, the strict constraints may amplify the impact of noise. In contrast, NAT does not explicitly require the model to output very rigid one-hot labels; instead, it induces the model to allocate certain probability outputs for similar classes (e.g., cats and dogs) and output smaller probabilities for less related categories (e.g., cats and airplanes) in a adaptive manner, thus learning robust features that generalize across categories. To validate this perspective, we compare NAT with the performance of these two regularizers, and the experimental results are shown in Table 11 where $\lambda_{\text{EMR}} = 0.001$ and $\lambda_{\text{NNR}} = 0.001$. Although EMR and NNR can bring certain increments to AT, NAT still demonstrates optimal results, proving that it cannot be simply replaced by methods that constrain output probabilities.

Furthermore, one may think about not constraining the model to learn one-hot output tendencies but rather allowing the model's output to approach a soft label. However, obtaining such soft labels is costly, especially when the desired suboptimal classes are similar to the target class. Such

Table 12: Evaluate NAT on person Re-Identification.

| Datasets | Defense | Clean | FNA | | SMA | | IFGSM | |
|---|---|---|---|---|---|---|---|---|
| | | | 8/255-16 | 10/255-16 | 8/255-16 | 10/255-16 | 8/255-16 | 10/255-16 |
| Market | None | 78.49/92.01 | 0.20/0.17 | 0.18/0.14 | 0.27/0.26 | 0.20/0.11 | 1.25/1.95 | 1.09/1.66 |
| | AMD | 69.69/88.24 | 8.57/18.14 | 4.37/9.41 | 22.85/35.69 | 15.21/23.37 | 17.97/34.65 | 11.74/23.34 |
| | AMD&NAT | **68.92/88.25** | **12.64/25.42** | **8.66/15.43** | **26.14/38.05** | **22.07/27.42** | **22.33/36.79** | **13.90/28.93** |

Table 13: Evaluate NAT on object detection.

| Attacks | Clean | $\text{loss}_{cls}$ | $\text{loss}_{loc}$ | DAG | RAP |
|---|---|---|---|---|---|
| Standard | 72.1 | 1.5 | 0.0 | 0.3 | 0.6 |
| MTD | 47.2 | 28.2 | 30.7 | 26.7 | 43.5 |
| MTD&NAT | **48.4** | **30.7** | **31.3** | **27.8** | **44.7** |

soft labels require extensive manual identification of which classes are "similar classes". Robustness distillation (Goldblum et al., 2020; Zhu et al., 2021; Zi et al., 2021) uses soft labels from the teacher's output to guide the student model. However, the reliability of the soft labels depends on the robustness of the teacher model. Reliable regularization is still needed to guide the training of the teacher model. An alternative regularization to obtain soft label is to use mixup (Zhang et al., 2017; 2020b) to generate augmented samples and soft labels. Compared to the adaptive allocation capability of NAT, mixup soft labels is manually set and do not guarantee that suboptimal classes are always categories similar to the target class. We also report the experimental results in Table 11, where the ratio of mixup is set to [0.7, 0.3] and randomly sampled in the current batch. Compared to mixup, NAT still demonstrates superiority.

Finally, we need to point out that although probability sparsity helps with adversarial robustness, directly applying sparsity regularization to vanilla training (which is actually the original nasty training (Ma et al., 2021)) does not achieve sufficient and reliable adversarial robustness. This is because such models can only learn sparse probabilistic outputs but still cannot acquire robust representations. The role of probability sparsity is to further guide the semantic interpretability and generalization of robust knowledge based on the model's learning of that robust knowledge. Results of NT in Table 11 validate the above viewpoint.

## L  PERFORMANCE ON NON-CLASSIFICATION TASKS

We further argue that NAT, through the orthogonalization of linear layers, can introduce a degree of feature space separation in the target model, and such property is transferable to other tasks. To further discuss the effectiveness of NAT for non-classification tasks, we consider two tasks distinct from pure classification: (1) Person Re-identification (classification + metric learning), and (2) Object detection (classification + regression).

First, for the person re-identification task, we adopt the common setup combining the cross-entropy loss and the triplet loss. We investigate commonly used adversarial metric attacks in ReID, including FNA Bouniot et al. (2020), SMA Bouniot et al. (2020), and IFGSM (e.g., AMA Bai et al. (2020)). The experimental results are shown in Table 12. The effectiveness of NAT for metric models trained with cross-entropy loss is foreseeable. As stated in the theoretical analysis, we argue that the probabilistic sparsity of NAT primarily stems from the high-order power constraint on the output logits, which can subsequently regularize the metric relationships in the feature space. Therefore, NAT can also bring performance gains to metric learning tasks.

Secondly, for the object detection task, we adopt the setup from Dong et al. (2022); Zhang & Wang (2019) and reproduce the results of Table 2 in Zhang & Wang (2019) on the PASCAL VOC dataset, based on which we further provide a preliminary validation of the efficacy of combining NAT with Multi-task Domain (MTD) training (since method in [3] is based on generative reconstruction, which is not conducive to the integration and fair comparison with NAT). As defined in [4], the MTD setup here refers to the defender performing min-max adversarial training by leveraging both the classification loss and the localization loss during training. We follow the standard NAT configuration in this experiment, applying the adversary model and NAT procedure only to the classification loss.

The attack methods employed include classification-loss-only attack ($\text{loss}_{\text{cls}}$), localization-loss-only attack ($\text{loss}_{\text{loc}}$), DAG Xie et al. (2017b), and RAP Li et al. (2018). The experimental results are shown in Table 13, which demonstrate that NAT not only brings robustness increments to the adversarial training framework based on the classification loss, but also generalizes to enhance the framework utilizing the localization loss. This outcome provides preliminary evidence that NAT can contribute to the adversarial robustness of object detection task.

In summary, we argue that NAT can stably bring robustness increments to adversarial training for non-classification tasks.

## M  LIMITATIONS

NAT can be regarded as a general improvement to commonly used adversarial, with its main contributions stemming from simple yet effective modifications to adversarial training and the exploration of robust sparsity attribution. The limitations of this work primarily lie in the evaluation and comparison on larger datasets (e.g., the full ImageNet-1k). On one hand, this is due to the massive computational complexity involved, and on the other hand, because the primary evaluation protocols within the adversarial robustness community have predominantly focused on datasets such as CIFAR, while benchmarks for ImageNet-1k remain relatively underdeveloped and establishing such a benchmark may be a huge task. Furthermore, extending our method to other downstream tasks could represent an interesting direction for future exploration. We leave these explorations for future work.

## THE USE OF LARGE LANGUAGE MODELS (LLMS)

LLMs are used sparingly and only to assist with proofreading and improving the linguistic fluency of a few sections of this paper (such as *Related Work*, *Experiments*, and *Appendix*), i.e., to aid or polish writing. All scientific contributions and core idea are the work of the authors. This assistance poses no issues of scientific ethics or misconduct.

