# OpenReview forum: "Nasty Adversarial Training:  A Probability Sparsity Perspective for Robustness Enhancement"
_ICLR.cc/2026/Conference — ICLR 2026 Poster_

### Official Review · Reviewer_AFqW · 2025-10-27

**Soundness:** 3
**Presentation:** 3
**Contribution:** 3
**Rating:** 6
**Confidence:** 3

**Summary:**

The paper proposed a method for improving adversarial training which incorporates nasty training. The method uses two networks: a target network which we would like to be more robust to adversarial example and a vanilla trained network. The target network is trained to be a "nasty teacher" version of the vanilla trained network. The intuition is that this keeps the top-1 prediction correct but sparse so top-N predictions are very low probability and therefore far away in the decision space of the model. This should make the model robust to perturbations of the input. The paper provides interesting theoretical analysis of the solution and empirical results show that it does work.

**Strengths:**

- Interesting and new idea: nasty training was intended for another purpose so its a good application
- Good analysis: the intuitive and formal analysis was interesting and mostly convincing
- Mostly good results: based on the results the method clearly does work

**Weaknesses:**

- The results seem to focus on CIFAR datasets which may not be fully representative of real world conditions (See Maiya et al. "Unifying the Harmonic Analysis of Adversarial Attacks and Robustness")
- No analysis of training time

**Questions:**

This was a very interesting paper which I think makes a good contribution. The idea is new and it's an insightful application of nasty training which makes sense from the theoretical motivation in the paper. The primary thing I think is missing is more/more convincing results. Based on the discussion I expected there to be a pretty clear improvement from the proposed method but on the presented CIFAR results, it didn't look like a dramatic change. Also I am not sure that CIFAR results are reflective of real conditions so the method should really be tested on something more comprehensive. It also it wasn't clear to me from the paper how much longer it takes to incorporate the proposed training loop after already going through adversarial training on the target model.

**Specific Questions**
- Does the method work on datasets beyond CIFAR? ImageNet for example?
- What is the total retraining time for this method? Both with and without the required adversarial training that occurs before the nasty training.

---

> ### Author Response · Authors · 2025-11-20
> **Thanks for the positive comments and the issues of concern to the reviewers have been answered**
>
> Thanks for the detailed review and feedback. Here are our responses to each questions.
>
> # 1.Evaluate NAT on ImageNet
>
> In the main text, we primarily evaluate NAT on the CIFAR dataset, as CIFAR serves as a common benchmark in the defense community and facilitates straightforward comparisons. Additionally, we have provided results of NAT executed on ImageNet-100 in Appendix B of the original version, demonstrating its effectiveness and superiority.
>
> # 2.Details of the time cost
>
> Our proposed NAT is essentially an end-to-end training method where Nasty Training and Adversarial Training are executed simultaneously within a unified process, with no requirement for Adversarial Training to precede Nasty Training. The time cost required for this training procedure has been provided in Appendix G of the original version. It is worth noting that the training time of the adversary model is not considered in Table 8. When the target model and the adversary model share the same architecture, the result of "vanilla training" in the first row can serve as a reference for the time cost of performing Vanilla Training (VT) on the adversary model. We have supplemented more detailed descriptions in the revised version; please refer to the Appendix H of the revised submission, where the content in blue font is supplementary.
>
> Thanks again for the positive comments!

---

### Official Review · Reviewer_gn7C · 2025-10-29

**Soundness:** 4
**Presentation:** 3
**Contribution:** 3
**Rating:** 6
**Confidence:** 4

**Summary:**

This paper introduces a method that integrate the nasty training (NT) into AT to strengthen robustness. They further analyze the probability sparsity of NT which has potential in adversarial training. And the experiments show the effectiveness of the method.

**Strengths:**

1. This paper analyzes the probability sparsity from NT and combined it with AT, by increasing decision boundary margins result in a higher cost for adversarial attacks, which shows the promising performance. And has in-depth theoretical interpretable analysis.
2. The experiment is comprehensive with CNN and ViT under different white and black attacks. And the ablations show the sparsity and effectiveness.

**Weaknesses:**

1. The main results in Table 1 and Table 2 were not compared with some SOTA method, such as LWTA [1], IKL-AT [2] and DCS [3].
2. All reported results correspond to the best outcomes over three independent runs, but there is no report of the mean and standard deviation of the results. And the class index in Figure 5 should be integer.

[1] Stochastic local winner-takes-all networks enable profound adversarial robustness, 2021.

[2] Decoupled kullback-leibler divergence loss, NeurIPS 2024.

[3] Adversarial Robustness via Deformable Convolution with Stochasticity, ICML 2025.

**Questions:**

1.	Could you add some up-to-date SOTA methods to make a more comprehensive comparison?
2.	Could you present the mean and standard deviation of your method at least in main table? And could you change the class index in Figure 5 to integer?
3.	NAT utilizes a VT for training. For the fairness, does NAT cover the expenses of the VT in Table 8?

---

> ### Author Response · Authors · 2025-11-20
> **Thanks for the suggestions and the issues of concern to the reviewers have been answered item by item**
>
> Thanks for the detailed review and feedback. Here are our responses to each questions.
>
> # 1.Compare NAT with the suggested SOTA defenses
>
> In the main text, we selected defenses that adhere to the standard experimental protocol for comparison. This includes evaluations against PGD, CW, and AutoAttack, demonstrating NAT's superiority under the same configuration.
>
> As suggested, we have added comparison experiments with IKL-AT [2] in Appendix E. Please refer to the updated Appendix E and Table 3 in the revised manuscript. Intuitively, NAT achieves state-of-the-art adversarial robustness while being easy to deploy and not significantly increasing computational complexity. Furthermore, it incurs less damage to the clean accuracy.
>
> However, the reviewer mentioned Refs. [1] and [3] are inherently random-structure based methods. These methods require evaluation under an adaptive attack framework which has not been provided.. Thus, it is unfair to directly compare these methods with our adversarial training-based method. However NAT can be easily integrated with random-structure based methods. When combined with random-structure based methods, can also achieve optimal robustness. .An initial experimental results are shown in the table below, where the adversary model still follows the standard setup, i.e., a naturally trained counterpart of the target model. A simple combination with NAT yields a significant and effective robustness increment.
>
> | widen factor | Natural Accuracy (%) |                     |                      | Robust Accuracy (%) |                     |                      |
> | :----------- | :------------------- | :------------------ | :------------------- | :------------------ | :------------------ | :------------------- |
> |              | **Baseline**         | **Stochasitc LWTA** | **Stochasitc LWTA & NAT** | **Baseline**        | **Stochasitc LWTA** | **Stochasitc LWTA & NAT** |
> | 1            | 74.01                | 87.0                | 88.4                 | 49.24               | 81.87               | 82.34                |
> | 5            | 83.95                | 91.88               | 92.01                | 54.36               | 83.4                | 84.42                |
> | 10           | 85.41                | 92.26               | 92.18                | 55.78               | 84.3                | 85.81                |
>
> [1] Stochastic local winner-takes-all networks enable profound adversarial robustness, 2021.
>
> [2] Decoupled kullback-leibler divergence loss, NeurIPS 2024.
>
> [3] Adversarial Robustness via Deformable Convolution with Stochasticity, ICML 2025.
>
> # 2.Correction of typographical errors and improvement of tables
>
> As suggested, we have supplemented the tables in the main text with the mean values and standard deviations of NAT. Additionally, we have corrected the clerical error in Figure 5. Please refer to Table 1, Table 2, and the revised Figure 5 in our revised submission. Thanks.
>
> # 3.Details of the time cost
>
> The time cost reported in Table 8 of the original version does not include the time cost of Vanilla Training (VT) for the adversary model, which is primarily based on the following considerations: (a) The architecture and training configuration of the adversary model are flexible and can generally contribute to robustness increments (as shown in Tables 6 and 7 of the original submission). (b) The time cost of VT is negligible compared to that of adversarial training. When the target model and the adversary model share the same architecture, the "vanilla training" result in the first row can serve as a reference for the VT time cost of the adversary model. We have added relevant explanations in the Appendix H of the revised version. Please refer to the revised Appendix H, where the content in blue font is supplementary.
>
> Thanks again for the positive comments!

---

### Official Review · Reviewer_2bUB · 2025-11-01

**Soundness:** 3
**Presentation:** 3
**Contribution:** 3
**Rating:** 6
**Confidence:** 3

**Summary:**

This paper introduces Nasty Adversarial Training (NAT), a defense framework for adversarial attacks. NAT integrates the concept of probability sparsity, which was originally proposed in the context of distillation resistance, into traditional adversarial training. The paper aims to demonstrate how enforcing sparsity in the output probability distribution can enhance the robustness of models by increasing the inter-class separability and widening decision margins in the logit space.

**Strengths:**

1. This paper use “nasty training” as a robustness regularizer.
2. This paper have solid theoretical reasoning and thorough experimental validation.
3. This method improves robustness while maintaining efficiency and interpretability.
4. This paper have good exposition of probability sparsity and its spatial interpretation.
5. This method can be easily integrated into existing AT pipelines with negligible cost.

**Weaknesses:**

1. The Taylor expansion and sparsity explanation rely on approximations; formal proofs or bounds are lacking.
2. The “spatial metric” benefits are qualitatively visualized but lack quantitative metrics (e.g., explicit margin distributions).
3. Only standard PGD/CW/AA considered — might not generalize to adaptive threat models.
4. The authors compare to entropy/logit norm regularization briefly but not in depth.
5. The term “nasty” may be unconventional for robustness literature and could obscure broader relevance.

**Questions:**

1. Can authors quantify the relationship between measured probability sparsity and empirical robustness (e.g., correlation between entropy and adversarial accuracy)?
2. How does NAT perform under adaptive attacks specifically designed to exploit the auxiliary adversary structure?
3. Could the “probabilistic sparsity” be approximated directly (e.g., via entropy regularization) instead of adversary-based NAT?
4. How sensitive is NAT to adversary model mismatch or overfitting? Would a partially shared backbone improve stability?
5. Does the spatial metric benefit persist for non-classification tasks (e.g., detection or segmentation)?

---

> ### Author Response · Authors · 2025-11-20
> **Thanks for the suggestions and the issues of concern to the reviewers have been answered item by item**
>
> Thanks for the detailed review and constructive feedback. Here are our responses to each questions.
>
> # 1.The relationship between measured probability sparsity and empirical robustness
>
> Quantifying a numerical relationship between probability sparsity and empirical robustness is indeed challenging. Even for the most fundamental metrics, such as the relationship between cross-entropy loss and classification accuracy, it is hard to derive a strict numerical correspondence. Similarly, establishing an exact numerical mapping between probability sparsity and robustness remains non-trivial. We primarily aim to demonstrate the positive effect of probability sparsity on robustness and to explain the origin of such sparsity.
> Still, a positive correlation between probability sparsity S and robustness R may be reasonably established. Probability sparsity S is intuitively connected to the minimum decision distance M. Empirically, M tends to increase as S increases, suggesting M ∝ S. Since the essence of adversarial attacks is to perturb an input across the decision boundary, the perturbation magnitude required for misclassification must exceed the minimum decision distance. Therefore, robustness R is directly related to M, i.e., R ∝ M. Combining the two relationships yields an intuitive overall trend: R ∝ M ∝ S.
>
> # 2. Evaluate NAT with adaptive attack
>
>  As suggested, we supplement the evaluation of NAT against adaptive attacks, where the adaptive attack incorporates the NT loss into the attack objective in addition to the standard classification loss. The results (Table 6 of the revised submission) demonstrate that NAT continues to maintain strong robustness even under adaptive attacks. Please refer to the Appendix D of the revised version for detailed discussion, with the supplementary content highlighted in blue.
>
>
> # 3.Directly using entropy regularization instead of NAT
>
> Directly applying entropy regularization may not be an optimal strategy. We have already provided comparisons with several existing regularization methods (including entropy-based regularization such as NNR) and presented the results in Appendix J of the original submission. Although these approaches can provide certain benefits, they still cannot achieve robustness gains comparable to those obtained by NAT. Please refer to the Appendix K of the revised version for detailed discussion.
>
>
> # 4.Discussion on the sensitivity of NAT to adversary model
>
> We have examined the effectiveness of NAT when the target model and the adversary model use different backbones (Appendix E, F, and H in the original submission). Results show that an architectural mismatch between the adversary model and the target model does not degrade robustness. This highlights the transferability of NAT across diverse adversary backbones. As a result, only sharing parts of the backbone may not yield additional performance gains, but adversary models with larger capacity  tend to produce a stronger regularization effect.
>
> # 5.Evaluate NAT on other non-classification tasks
>
> As suggested, we consider two tasks distinct from pure classification: (1) Person Re-identification (metric learning), and (2) Object detection (regression task). The segmentation task shares certain commonalities with the detection task.The experimental results of NAT on ReID (Table 12 of the revised submission) shows that NAT can bring robustness gains to metric learning tasks. Also, the results of NAT on objective detection (Table 13 of the revised submission) demonstrate that NAT also generalizes to enhance the framework including localization task.  Thus, we argue that NAT can stably bring robustness gains for non-classification tasks. Please refer to Appendix L in the revised submission for detailed settings and discussions, where the content in blue font is supplementary.
>
> [1]Bouniot Q, Audigier R, Loesch A. Vulnerability of person re-identification models to metric adversarial attacks[C]//Proceedings of the IEEE/CVF conference on computer vision and pattern recognition workshops. 2020: 794-795.
>
> [2]Bai S, Li Y, Zhou Y, et al. Adversarial metric attack and defense for person re-identification[J]. IEEE Transactions on Pattern Analysis and Machine Intelligence, 2020, 43(6): 2119-2126.
>
> [3]Dong Z, Wei P, Lin L. Adversarially-aware robust object detector[C]//European Conference on Computer Vision. Cham: Springer Nature Switzerland, 2022: 297-313.
>
> [4]Zhang H, Wang J. Towards adversarially robust object detection[C]//Proceedings of the IEEE/CVF International Conference on Computer Vision. 2019: 421-430.
>
> Thanks again for the constructive feedback and positive comments!

---

### Author Response · Authors · 2025-12-01
**All the comments have been addressed, and the manuscript has been updated to standardized version.**

We have addressed all the reviewers' comments and have incorporated all of their suggested experiments and modifications, which primarily include:

(1) Discussion on the relationship between measured probability sparsity and empirical robustness;

(2 )Discussion on the adaptive attack;

(3) Comparison between NAT and Entropy Regularization;

(4) Discussion on the sensitivity of NAT to adversary model;

(5) Evaluate NAT on other non-classification tasks;

(6) Further evaluation and comparison of NAT with recommended defense methods and datasets;

(7) Details of the time cost;

(8) Some typos.

Thank you all !

Supplement: Due to the approaching deadline, we have updated the current submission. The modifications and additions previously highlighted in blue have now been standardized to black font. For specific changes, please refer to our responses to the reviewers' comments. Thanks again !

---

### Meta-Review · Area_Chair_5iwC · 2026-01-07

**Summary:**

Reviewer **2bUB** acknowledges the novelty of this paper and recognizes its theoretical analysis. The main concerns come from the lack of formal proofs. The authors did not respond to those concerns.

Reviewer **gn7C** acknowledges the theoretically interpretable analysis of the proposed method. The main concerns come from the lack of comparison with the SOTA methods. The reviewer suggested three SOTA methods, and the authors compared them with one of them.

Reviewer **AFqW** recognizes the interesting theoretical analysis and the good empirical results of this paper. The reviewer suggests evaluating more datasets beyond the CIFAR datasets. The authors point out that Appendix B of the original version already includes the results on more datasets.

**Reviewer Concerns:**

I think most of the concerns have been addressed. The only one remaining is the comparison with more SOTA methods.

**Reviewer Scores:**

I think most of the reviewers will keep their positive rating.

---

### Decision · Program_Chairs · 2026-01-26

Accept (Poster)